# Real-Time Micro-Monitoring of Surface Temperature and Strain of Magnesium Hydrogen Tank through Self-Made Two-In-One Flexible High-Temperature Micro-Sensor

**DOI:** 10.3390/mi13091370

**Published:** 2022-08-23

**Authors:** Chi-Yuan Lee, Chia-Chieh Shen, Chun-Wei Chiu, Hsiao-Te Hsieh

**Affiliations:** Department of Mechanical Engineering, Yuan Ze Fuel Cell Center, Yuan Ze University, Taoyuan 32003, Taiwan

**Keywords:** two-in-one flexible high-temperature micro-sensor, magnesium hydrogen tank, real-time micro-monitoring

## Abstract

The adsorption and desorption of hydrogen in the magnesium powder hydrogen tank should take place in an environment with a temperature higher than 250 °C. High temperature and high strain will lead to reactive hydrogen leakage from the magnesium hydrogen tank due to tank rupture. Therefore, it is very important to monitor in real time the volume expansion, temperature change, and strain change on the surface of the magnesium hydrogen tank. In this study, the micro-electro-mechanical systems (MEMS) technology was used to innovatively integrate the micro-temperature sensor and the micro-strain sensor into a two-in-one flexible high-temperature micro-sensor with a small size and high sensitivity. It can be placed on the surface of the magnesium hydrogen tank for real-time micro-monitoring of the effect of hydrogen pressure and powder hydrogen absorption expansion on the strain of the hydrogen storage tank.

## 1. Introduction

Taiwan plans to make renewable energy account for 20% of its overall power generation by 2025, so Taiwan has and will have a very high demand for large and long-term energy storage equipment at present and in the future. At the 26th session of the Conference of the Parties to the United Nations Framework Convention on Climate Change (Conference of the Parties, COP26), many countries made new commitments. For example, more than 50 countries and the entire European Union pledged to achieve the target of net-zero emissions; meanwhile, they pledged to achieve the goal that the annual new installed capacity of solar and wind energy will be close to 500 GW by 2030. The conference also pointed out that more and more countries had formulated policies related to hydrogen energy and set goals for developing hydrogen energy technology, while emphasizing the role that hydrogen and fuel cell technology can play in the transformation of global clean energy. Magnesium hydrogen tanks will shrink and expand due to different states of hydrogen absorption and desorption, so it is necessary to measure the volume expansion, temperature change, and strain change of the hydrogen tank. Compressed gas storage is the most recognized type of hydrogen storage. The internal pressure must be increased to 70 MPa to meet industry standards, such as gravimetric and volumetric hydrogen storage capacity [1,2,3,4,5].

Hydrogen is considered one of the most attractive future energy carriers. Hydrogen is mostly stored in the solid state at present. Solid state storage employs the methods of using physical or chemical adsorption to store hydrogen. Magnesium hydride is considered a suitable material for hydrogen storage applications, due to a hydrogen content of 7.6 wt.% associated with hydrogenation/dehydrogenation reaction enthalpy of about 74 kJ/molH_2_, and to a high degree of reversibility. The adsorption and desorption of hydrogen in the magnesium hydrogen tank should take place in an environment with a temperature higher than 250 °C. High strain will lead to reactive hydrogen leakage from the magnesium hydrogen tank due to tank rupture [6,7,8].

The purpose of this study was to develop a micro-strain sensor with high temperature resistance and integrate micro-temperature sensors using micro-electro-mechanical systems (MEMS) technology, so as to realize the real-time micro-monitoring of the effect of hydrogen pressure and powder hydrogen absorption expansion on the strain of the hydrogen storage tank.

According to the previously proposed literature [9], the photomask was modified and successfully tested for 120 h to explore the effect of the hydrogen absorption expansion of the powder on the strain of the hydrogen storage tank.

## 2. Literature Review

Otsuki et al. [10] developed a thin film thermocouple (TFTC), by which they sputtered Au and Ni on the parylene thin film and used parylene as an insulating layer with an overall thickness of only 7 μm and accuracy of ±0.1 °C. This micro-sensor can be embedded in a proton exchange membrane fuel cell (PEMFC) for measurement. 

Kang et al. [11] created RTD (resistance temperature detector) temperature sensors on polyimide (PI), using Ag as the sensing material. 

Sarajlić et al. [12] proposed a temperature sensor based on four thin film resistors which meandered on the surface of the silicon chip in the form of thin metal layers. Specifically, two resistors were covered with a layer of 2.3 µm hard-baked photoresist, and the other two were exposed to ambient air. 

Morisaki and Zhou et al. [13,14] made two aluminum RTDs on the top of SiO_2_ substrates and coated 2 µm of SU-8 between the two aluminum RTDs as an insulating layer, making the aluminum RTDs a set of heat flux sensors. 

Cui et al. [15] used nickel (Ni) to make a resistance temperature sensor. He deposited a Ni thin film on an alumina substrate with a sputtering machine to make a sensor and tried to deposit it at different temperatures during the sputtering process; he also checked the state of the Ni thin film by testing equipment such as XRD and SEM. In the literature on micro-strain sensors, Ottermann et al. [16] proposed that a sputtered thin film metal strain gauge with small thicknesses could provide new measurement locations in harsh environments. 

Shu et al. [17] adopted a low-cost PU film (50 µm) that exhibits good biological compatibility and is commonly used in clinic. After the substrate parts were prepared, the flexible sensor was produced by sputtering 30 nm thickness Au film as the sensitive and the electrode layers on PU’s surface.

Kal et al. [18] proposed that electroplated gold film has the advantages of anti-oxidation, low resistance, good overall chemical inertness, etc.

## 3. Design of a Two-In-One High-Temperature Flexible Micro-Sensor

### 3.1. Design Principle of the Micro-Temperature Sensor

Gold is characterized by Positive Temperature Coefficient (PTC), and its resistivity increases when its temperature rises; this characteristic results from the temperature coefficient of resistance (TCR) of the conductor. Au is used in the device [19]. As the environmental temperature increases, the resistance of the resistance temperature sensor (RTD) also increases, because a metal conductor has a PTC. The micro-temperature sensor developed in this study is an RTD, as shown in Figure 1. The sensing principle of the micro-temperature sensor is that when the ambient temperature rises, due to the PTC characteristic of gold, the resistance value increases as the temperature rises. This characteristic is caused by the TCR of the conductor. The linear equation between resistivity (*ρ*) and temperature (*T*) is defined as Equation (1), where *ρ* is the resistivity at temperature *T*, *ρ*_0_ is the resistivity at the reference temperature (*T*_0_), and α is the temperature coefficient of resistance TCR. Gold (Au) with good physical and chemical properties and a simple process was used as the electrode material. In order to increase the basic resistance and facilitate the measurement of changes, the electrode was designed as a serpentine structure with an electrode area of 400 µm × 400 µm. The minimum line breadth of the electrode was 10 µm, the line spacing was also 10 µm, and the basic resistance at 25 °C was 1000 Ω ± 10%, thickness 25 µm.
(1)$$\alpha =\frac{1}{\rho_{0}}\frac{d\rho }{dT}$$

### 3.2. Design Principle of the Micro-Strain Sensor 

Strain refers to the change ratio of the length of the object after a force to the original length without a force, as shown in Figure 2. Strain can be divided into four different types: axial, bending, shear, and torsion. Although the surface of the hydrogen tank used in this study was an arc, it was regarded as a linear axial stretch because of the small size of the strain sensor. Generally, a metal strain sensor must include metal sheets arranged in a grid pattern, and the grid pattern is a serpentine electrode. This arrangement can maximize the length of the metal line, amplify the basic resistance, and improve the sensitivity, which can facilitate correction and micro-monitoring. In this study, gold (Au) was also used as the material of the micro-strain sensor with an electrode area of 400 µm × 400 µm. The minimum line breadth of the electrode was 10 µm, the line spacing was also 10 µm, and the basic resistance at 25 °C was 1000 Ω ± 10%, thickness 25 µm.

## 4. Production Process of a Two-In-One High-Temperature Flexible Micro-Sensor

### 4.1. Polyimide Thin Film Cleaning

Polyimide has the advantages of high temperature resistance, chemical corrosion resistance, and high strength. The polyimide thin film was cut into a circle with a diameter of 10 cm. Then, it was carefully cleaned to ensure the cleanness of the substrate surface. The first step was to soak the substrate in acetone and shake it in an ultrasonic shaker. The second step was to use isopropanol to clean the residual acetone on the substrate. The third step was to clean the substrate with deionized water. The last step was to use a pressurized air gun to blow off the residual water on the surface of the substrate; the substrate was placed on a heating plate to bake at 100 °C, and the cleaned polyimide thin film was pasted on the glass test piece to facilitate subsequent processes. The polyimide thin films used in this study were all processed through the corona treatment when purchased to increase the metal adhesion. Corona treatment refers to using high-energy ions to bombard the surface of the material, thus destroying the molecular structure on the surface of the material and causing the surface of the material to be roughened.

### 4.2. Lithography—Definition of an Electrode Pattern

In the MEMS process, lithography is the most important, critical, and indispensable step. The success of the entire process depends largely on the smoothness of the lithography. The electrode pattern is defined by the use of positive photoresist AZ^®^ P4620. After the photoresist is coated with a thickness of about 10 μm using a spin coater, the coated photoresist should be baked on top of a heating plate to evaporate the excess solvent in order to shape the photoresist. This step is called soft bake. The next step is exposure. In this study, a double-sided mask aligner and exposure system (AG-200-4N-D-SM, M & R Nano Technology Co., Taoyuan City, Taiwan) was used for exposure, and the designed mask was properly placed. Generally, a post-exposure bake (PEB) process is required after photoresist exposure, but this type of photoresist can be developed without PEB. After exposure, development processing is performed. The development process is to transfer the mask pattern to the photoresist that is presented by chemical reaction, and the previous photomask is modified to facilitate the increase of yield during the exposure process, as shown in the photo of the mask in Figure 3. The developer used in this study was the mixture of AZ^®^ 400 K and deionized water (DI Water). The exposed test piece was soaked into the developer. During the soaking process, the surface of the test piece was rinsed with a dropper to help to develop. The developed graphic photo is shown in Figure 4.

### 4.3. Metal Thin Film Deposition

Electron beam evaporation was used in this study for metal thin film deposition, which belongs to physical vapor deposition (PVD). Two-in-one flexible high-temperature micro-sensors are all deposited with gold (Au) as the sensing layer. Since in the lift-off process, evaporation comes after photoresist coating, the metal in the pattern after development is smooth and flat, while the areas still covered by photoresist remain rough and uneven. After evaporation, the test piece can be lifted off immediately. In the lift-off process of this study, acetone was used to remove the original photoresist and the excess metal was lifted off at the same time, leaving only the metal of the electrode pattern on the test piece. First, the test piece was soaked in acetone and placed in an ultrasonic oscillator for vibration until there was no large area of metal residue.

### 4.4. Lithography Protective Layer

The protective layer was made of Fujifilm Electronic Materials U.S.A., Inc. LTC^®^ 9320 liquid polyimide with high mechanical strength, anti-corrosion, and stretchability, which can be resistant to an electrochemical environment. First, a spin coater was used for coating, and then the material was placed on a heating plate for baking after the coating was completed. The exposure came after soft baking. This type of liquid polyimide needs to be baked after exposure and also needs to be placed on a heating plate for baking. The exposure and baking can make the exposed pattern appear and make the overall pattern perfect by eliminating the standing wave effect of light at the exposed position. After the exposure was finished, the developer could start to develop after the temperature is lowered. After the protective layer was developed without any problem, the test piece was put into a constant temperature oven (DENG YNG^®^ DS45) to bake. The liquid polyimide was cured after high-temperature baking, and the cured protective layer can resist corrosion, scratch, and wear. The photo after all processes were completed is shown in Figure 5.

## 5. Magnesium–Titanium Hydrogen Storage Tank

The length of the hydrogen storage tank is 80 mm, and the inner diameter is 16 mm. The metal hydride powder used in the hydrogen storage tank weighs 4.355 g, and the ratio of magnesium hydride to titanium hydride is 9:1. The magnesium powder absorbs hydrogen. The reaction formula is as follows:Mg + H_2_ → MgH_2_ + (thermal energy)(2)

## 6. Correction and Micro-Monitoring of Two-In-One Flexible Micro-Sensor

### 6.1. Instant Microscopic Monitoring of Surface Temperature and Strain of Hydrogen Storage Tank

The high-temperature flexible two-in-one micro-sensor was placed on the surface of the hydrogen storage tank to test the data of real-time micro-monitoring of the temperature and strain of a single hydrogen desorption reaction. The overall reaction time was about 3 h and 30 min. It will absorb heat, so it was originally expected that the surface temperature of the hydrogen storage tank should decrease. In fact, because the thermocouple received the cooling signal, the power of the high-temperature furnace was immediately increased, so the surface temperature of the hydrogen storage tank was kept stable at around 250 °C; the strained part also maintained at around 0.013ε, as shown in Figure 6.

Then, the high-temperature flexible two-in-one micro-sensor was placed on the surface of the hydrogen storage tank to continuously test 30 hydrogen absorption and desorption cycles. A single hydrogen absorption and desorption cycle took about 4 h, and the total time of 30 cycles was about 120 h. The measured data are the maximum strain of each cycle to draw a curve (Figure 7). This continuous 30-cycle test verifies that the high-temperature flexible two-in-one micro-sensor developed in this research can be in continuous use in this high-temperature environment for more than 120 h.

### 6.2. Magnesium–Titanium Hydrogen Storage Tank Thermal Expansion Test

Since the strain curve measured in the experiment was consistent with the temperature change, an experiment was designed to verify whether it was directly related to the temperature. The experimental design and steps are as follows:

(a) Prepare a hydrogen storage tank without storing hydrogen storage powder.

(b) Paste the micro-sensor in the same position in the same way.

(c) The temperature is set at 250 °C, the hydrogen is directly passed into the hydrogen storage tank, and the target pressure is set at 1.0 MPa.

(d) Arrange the measured data and calculate the dependent variable, as shown in Figure 8.

(e) The hydrogen is released from 1.0 MPa to a pressure of 0.01 MPa in the tank, and the strain is also calculated as shown in Figure 9.

(f) Finally, maintain the pressure in the tank at 0.01 MPa, heat the temperature from 250 °C in 10 °C increments, hold each temperature for 3 min to wait for stabilization, and capture data for 3 min until the temperature reaches 290 °C. Take the average value of the resistance value for each temperature value, and draw a curve as shown in Figure 10. Calculate the strain after deducting the temperature compensation from the resistance value, as shown in Figure 11.

## 7. Conclusions

In summary, the temperature of the micro-temperature sensor is about 252 °C, and the strain is about 0.014 to 0.015ε during hydrogen charging, and the temperature of the micro-temperature sensor is about 248 °C and the strain is about 0.013 to 0.015ε when hydrogen is discharged. Floating within 0.014ε, the strain of an empty tank under 1 atm is 0.0135ε at 250 °C, and the error value is 2–4% of the reading value, so the strain caused by hydrogen pressure cannot be seen in the measurement data. The relationship between Young’s modulus (E) and strain is shown in Formula (2). While the Young’s modulus of stainless steel is 193 GPa, and ε is 0.00000518 calculated by substituting σ = 1.0 MPa into Formula (3), it can be known that the amount of strain caused by the gas pressure is already smaller than the error value of the measuring equipment, so it can be assumed to be 0.
E = σ/ε(3)

Finally, comparing Figure 6 with Figure 8, it can be seen that the strain amount of each temperature value is the same as the measured value of the empty tank without hydrogen storage powder. Therefore, it can be concluded that when hydrogen is absorbed and desorbed, regardless of whether there is hydrogen storage powder, the pressure is set when the pressure is set. Below 1 MPa, the hydrogen pressure and powder hydrogen absorption expansion did not cause strain on the hydrogen storage tank.

## Figures and Tables

**Figure 1 micromachines-13-01370-f001:**
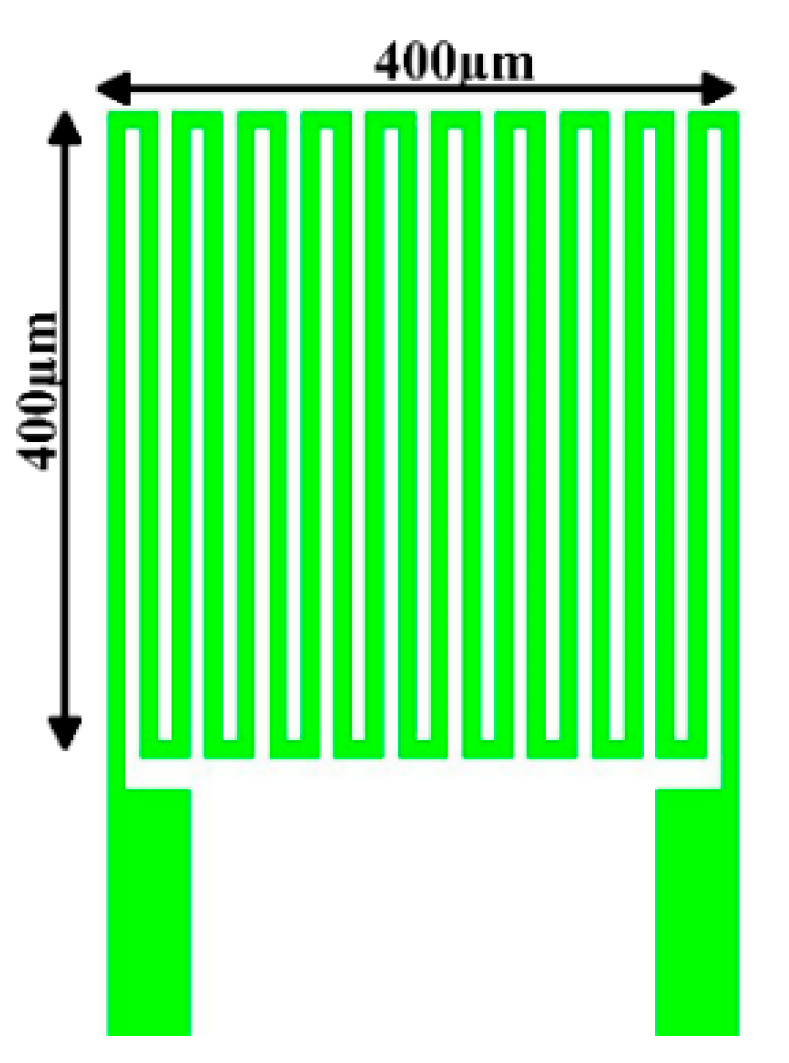
Principle and design of micro-temperature sensor.

**Figure 2 micromachines-13-01370-f002:**
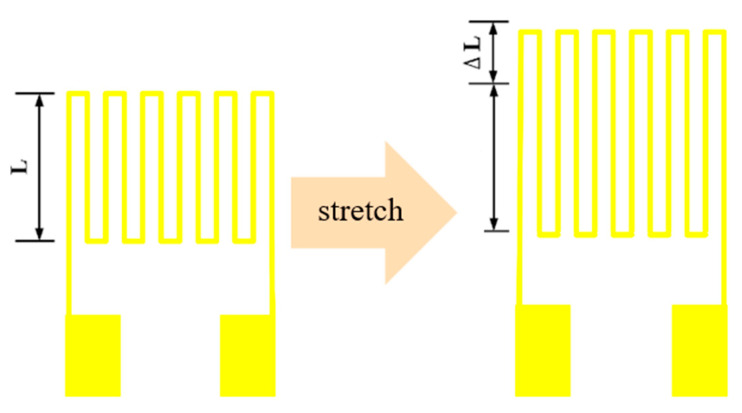
Principle and design of micro-strain sensor.

**Figure 3 micromachines-13-01370-f003:**
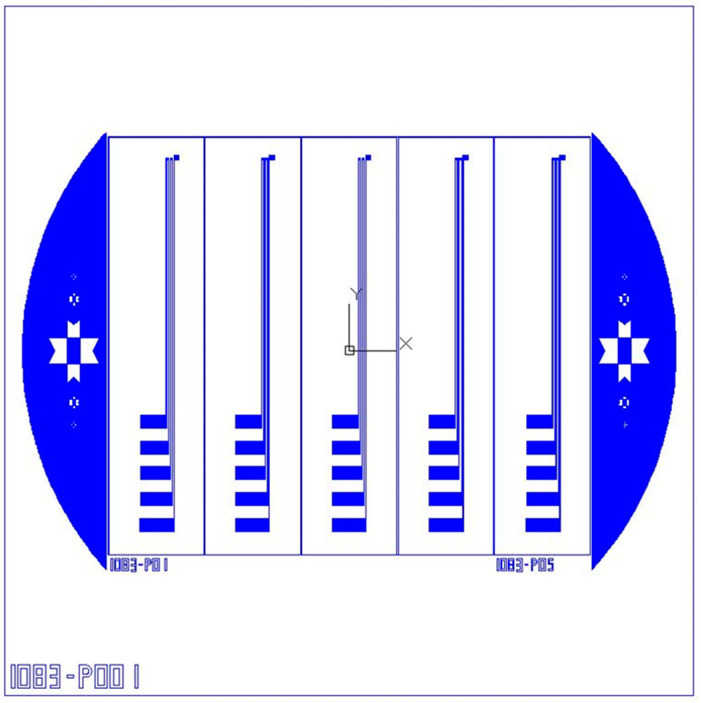
Mask photo.

**Figure 4 micromachines-13-01370-f004:**
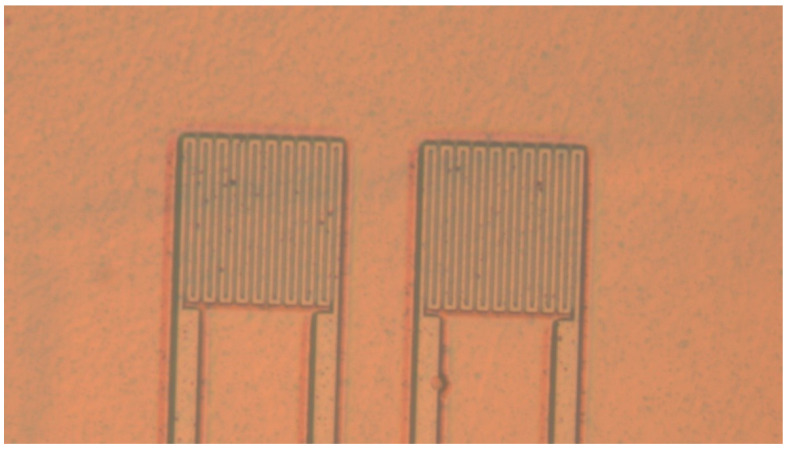
Electrode pattern after development.

**Figure 5 micromachines-13-01370-f005:**
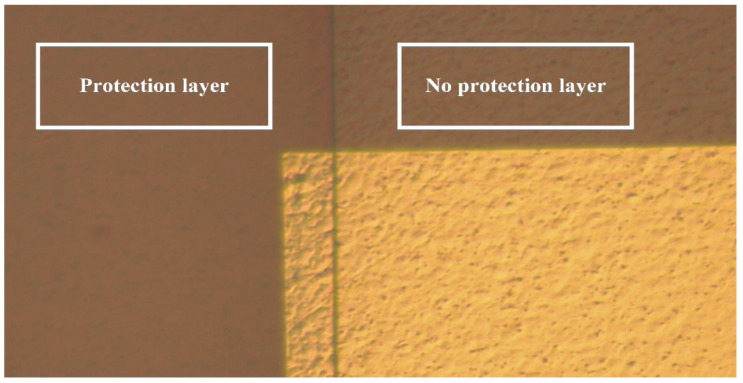
Optical microscope photo of the protective layer.

**Figure 6 micromachines-13-01370-f006:**
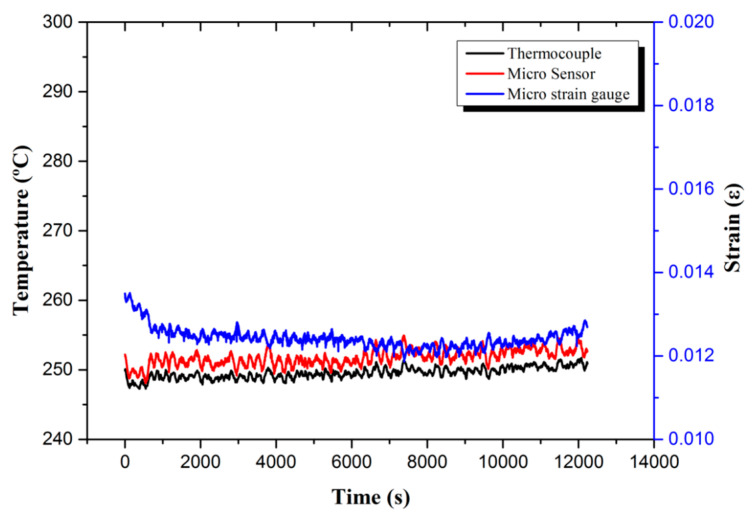
Real-time micro-monitoring data of temperature and strain of a single hydrogen absorption reaction.

**Figure 7 micromachines-13-01370-f007:**
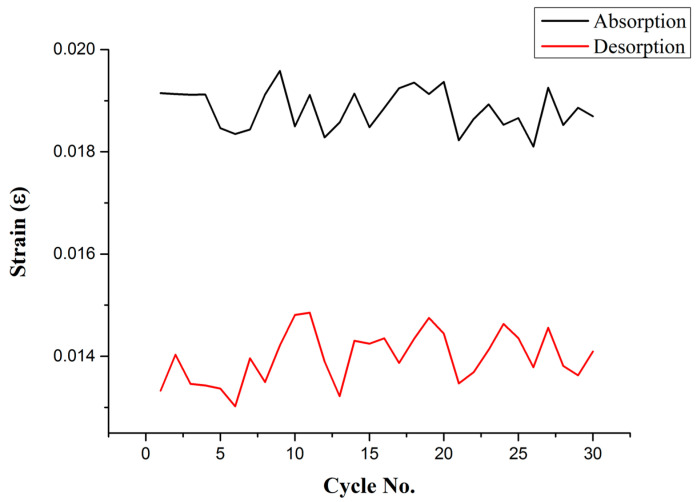
Maximum strain value of hydrogen storage tank for 30 cycles of hydrogen absorption and desorption.

**Figure 8 micromachines-13-01370-f008:**
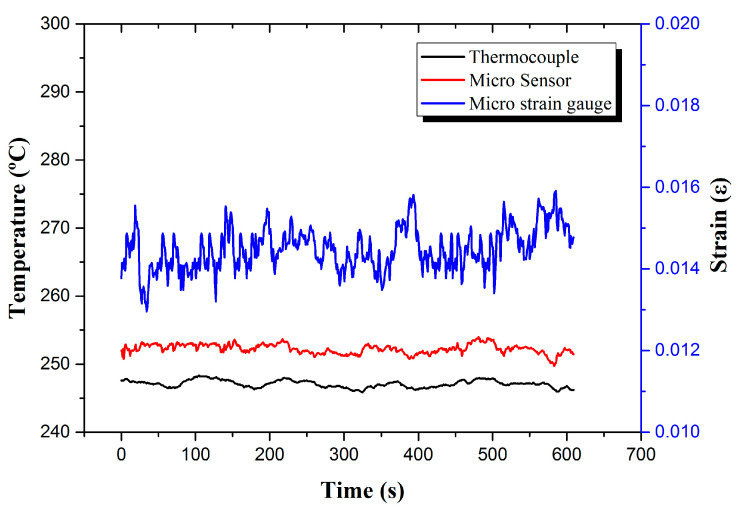
Real-time micro-monitoring of hydrogen charging temperature and strain of hydrogen storage tank.

**Figure 9 micromachines-13-01370-f009:**
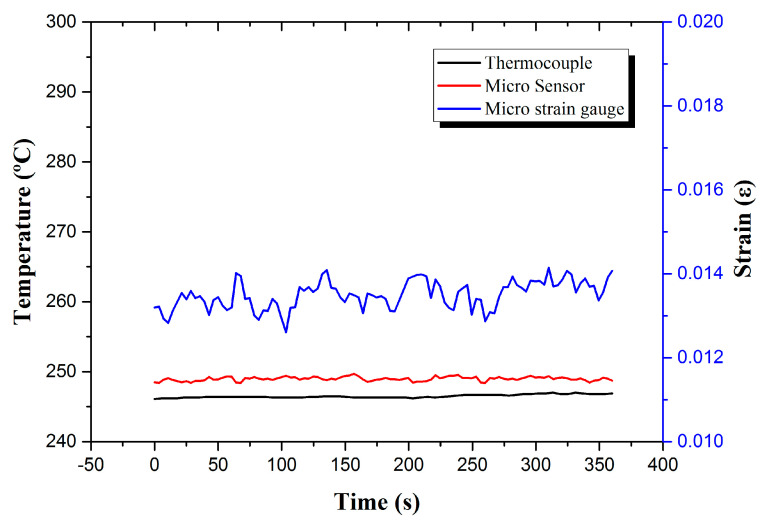
Real-time micro-monitoring of hydrogen desorption temperature and strain of hydrogen storage tank.

**Figure 10 micromachines-13-01370-f010:**
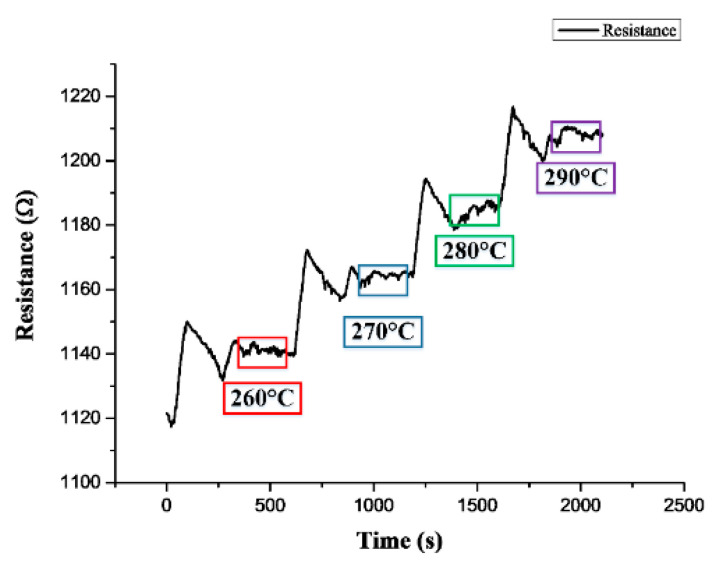
Resistance value measured by micro-strain sensor at different temperatures of hydrogen storage tank.

**Figure 11 micromachines-13-01370-f011:**
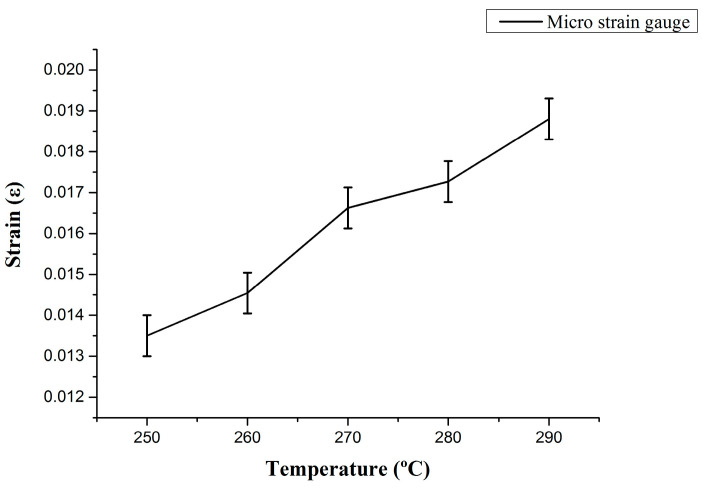
The strain value measured by the micro-strain sensor at different temperatures of the hydrogen storage tank.

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
