# Peer review of "Real-Time Micro-Monitoring of Surface Temperature and Strain of Magnesium Hydrogen Tank through Self-Made Two-In-One Flexible High-Temperature Micro-Sensor"

_micromachines, 2022, doi:10.3390/mi13091370_

Round 1
Reviewer 1 Report
Some new information is positive, example in the lines 55-57 the authors expose:
“According to the previously proposed literature [9], the photomask was modified and successfully tested for 120 hours to explore the effect of the hydrogen absorption ex pansion of the powder on the strain of the hydrogen storage tank”
COMMENT “However some details are unclear in this paragraph. Because the authors expose the “photomask” was modified and tested, maybe the authors are addressing to a new pattern for the fabricated sensor. The photomask is only the device for transfering the patterns.
The main controversy is in the lines 246 – 256, where the authors expose the following:
“In summary, the temperature of the micro-temperature sensor is about 252°C and the strain is about 0.014 to 0.015ε during hydrogen charging, and the temperature of the micro-temperature sensor is about 248°C and the strain is about 0.013 to 0.015ε when hydro gen is discharged. Floating within 0.014ε, the strain of an empty tank under 1 atm is 0.0135ε at 250°C, and the error value is 2%-4% of the reading value, so the strain caused by hydrogen pressure cannot be seen in the measurement data; material Yang The relationship between Young's modulus (E) and strain is shown in formula (2), while the Young's modulus of stainless steel is 193GPa, and ε is 0.00000518 calculated by substituting σ=1.0MPa into formula (2) , it can be known that the amount of strain caused by the gas pressure is already smaller than the error value of the measuring equipment, so it can be assumed to be 0.”
COMMENT: These asseverations are unclear including the 2-4% error value, because watching the graphs at Figure 11, for the strain the margin of variation is 0.01, the measured strain range is 0.0135 – 0.0190, then the error value is unclear. Similar details are evidenced in Fig. 8 for strain scale
The main question is: how is the error value calculated?
GENERAL COMMENTS:
1 Some authors reply are missing because they only erase the questioned paragraphs or make unclear comments. For example see the authors reply, in the first paragraph
2 Along the new paper version, several changes are unidentified because the modifications was made without some mark
3 In the new file some typos are identified
Reviewer 2 Report
Your 2 in 1 micro sensor has a unique design structure and has good performance to measure temperature and strain at a high temperature of 250 degrees Celsius. The following corrections and additional explanations are necessary for the completeness of your paper.
(1) For the completeness of your paper, the following minor corrections are necessary.
- Lines 87~96: The full names of PTC, TCR, and RTD are repeated a lot. Abbreviations once explained are unnecessary to express their full names.
- Eq (1): Enter a description of the parameter.
- Figure 6, 8, 9: I am confused which curve is Temperature or Strain. Show the temperature uniformly with a black line (Themocouple: solid line, Micro Sensor: dotted line) and strain with a blue line.
- Figure 8, 9: The picture is not clear. Please correct it.
- line 206: the total time of 30 cycles was about 4 hours. -> Check whether it is 4 hours or 120 hours.
(2) About 2 in 1 sensor
- Your 2 in1 micro sensor is amazing. However, the explanation of the compensation part of the mutual interference between the strain sensor and the temperature sensor is insufficient.
When the temperature rises, it affects the strain sensor, and when the strain increases, it affects the temperature sensor.
A methodological explanation is needed to correct this.
- The size of the hydrogen tank used in the test is very small. In addition, the tested pressure is also a relatively small value of 1 MPa. It is obvious that this hydrogen tank has a small strain value.
What is the actual strain value of the surface of the industrial hydrogen tank, and does your micro strain sensor have a measurable range?
- Figure 9: There is no relationship between tank pressure and strain. How does the measured stain change when the tank pressure increases?

Round 2
Reviewer 1 Report
Some authors' reply is still slightly unclear. In the conclusions section, line 251, what means: material Yang?
However, the paper looks like enough work.
Author Response
Please see the attached file.

This manuscript is a resubmission of an earlier submission. The following is a list of the peer review reports and author responses from that submission.
Round 1
Reviewer 1 Report
Real-time micro-monitoring of surface temperature and strain of magnesium hydrogen tank through self-made two-in-one flexible high-temperature micro-sensor
Chi-Yuan Lee *, Chia-Chieh Shen, Chun-Wei Chiu and Hsiao-Te Hsieh
Department of Mechanical Engineering, Yuan Ze Fuel Cell Center, Yuan Ze University, Taoyuan, Taiwan, R.O.C.
The authors expose the following:
“At present, the temperature sensors and strain gauges that are commercially purchased are large, so it is difficult to place the two sensors on the surface of magnesium hydrogen tanks for accurate measurement at the same time”
Lines 64-66
The authors do not mention the precise available room for the sensor devices.
About the asseveration:
“The purpose of this study was to develop a micro-strain sensor with high temperature resistance and integrate micro-temperature sensors by using micro-electro-mechanical systems (MEMS) technology, so as to realize the real-time micro-monitoring of the surface temperature, expansion, and strain change of the magnesium hydrogen tank…”
Lines 101 - 106
Several steps are presented under very general considerations, for example:
The micro-temperature sensor, 25°C, 1000W±10%, shown in Fig. 1.
The micro-strain sensor, 25°C, 1000W±10%, shown in Fig. 2
Both devices are presented without some discussion of the metal thickness and sizing. Additionally, the W±10% tolerance is too much wide for the mentioned application
Regarding the thermal conductivity issue, the glue and the polyimide films, both for the substrate and capping, were utilized without technical consideration.
Moreover, the windows opening onto the metal contacts and additional wiring are missing in the experimental description; those procedures are critical and fundamental for the mentioned application. In this sense, both images in Figs 3 and 4 are unclear and incomplete. Finally, section 4 is unclear by the details described previously
In the conclusions section, the authors expose:
“… stable, and correction tests in different simulated environments demonstrated its high linearity and reliability”
Lines 253 and 254
Authors must clarify this asseveration in the experimental section.
Additionally:
The Keywords are wrong because keywords are not descriptive phrases.
Examples could be: Hydrogen tanks, temperature sensors, strain sensors, MEMS, flexible electronics, polyimide . . . and so on.
In the introduction section, gold films lack a rigorous discussion as an alternative material for the sensors, .only Ref [12] is mentioned.
Some sections are too long and technically unclear, for example, lines 171 - 185 (metal evaporation).
Reviewer 2 Report
This manuscript is very similar to or even the same as the below publication from the same corresponding author. Therefore, I reject it.
Chi-Yuan Lee, Chia-Chieh Shen, Shuo-Jen Lee, Chun-Wei Chiu, Hsiang-Ting Lin,
Real-time microscopic monitoring of temperature and strain on the surface of magnesium hydrogen storage tank by high temperature resistant flexible integrated microsensor,
International Journal of Hydrogen Energy,
Volume 47, Issue 25,
2022,
Pages 12815-12821,
ISSN 0360-3199,
https://doi.org/10.1016/j.ijhydene.2022.02.003.